nanotechnology/cellular biology

magnetic Fe₃O₄ nanoparticles, cytotoxic effects, oxidative index, apoptosis, chicken macrophage cells

**Author for correspondence:**
Shourong Shi
e-mail: ssr236@163.com

This article has been edited by the Royal Society of Chemistry, including the commissioning, peer review process and editorial aspects up to the point of acceptance.

# Cytotoxicity studies of Fe₃O₄ nanoparticles in chicken macrophage cells

Shan Zhang[1], Shu Wu[1], Yiru Shen[1], Yunqi Xiao[1], Lizeng Gao[2] and Shourong Shi[1,3]

[1]Poultry Institute, Chinese Academy of Agricultural Sciences, Yangzhou, Jiangsu 225125, People's Republic of China
[2]Institute of Biophysics, Chinese Academy of Science, CAS Engineering Laboratory for Nanozyme, Institute of Biophysics, CAS, Beijing 100101, China
[3]Jiangsu Co-innovation Centre for Prevention and Control of Important Animal Infectious Diseases and Zoonoses, Yangzhou, Jiangsu 225000, People's Republic of China

SZ, 0000-0001-5823-3112

Magnetic Fe₃O₄ nanoparticles (Fe₃O₄-NPs) have been widely investigated for their biomedical applications. The main purpose of this study was to evaluate the cytotoxic effects of different sizes of Fe₃O₄-NPs in chicken macrophage cells (HD11). Experimental groups based on three sizes of Fe₃O₄-NPs (60, 120 and 250 nm) were created, and the Fe₃O₄-NPs were added to the cells at different doses according to the experimental group. The cell activity, oxidative index (malondialdehyde (MDA), superoxide dismutase (SOD) and reactive oxygen species (ROS)), apoptosis and pro-inflammatory cytokine secretion level were detected to analyse the cytotoxic effects of Fe₃O₄-NPs of different sizes in HD11 cells. The results revealed that the cell viability of the 60 nm Fe₃O₄-NPs group was lower than those of the 120 and 250 nm groups when the same concentration of Fe₃O₄-NPs was added. No significant difference in MDA was observed among the three Fe₃O₄-NP groups. The SOD level and ROS production of the 60 nm group were significantly greater than those of the 120 and 250 nm groups. Furthermore, the highest levels of apoptosis and pro-inflammatory cytokine secretion were caused by the 60 nm Fe₃O₄-NPs. In conclusion, the smaller Fe₃O₄-NPs produced stronger cytotoxicity in chicken macrophage cells, and the cytotoxic effects may be related to the oxidative stress and apoptosis induced by increased ROS production as well as the increased expression of pro-inflammatory cytokines.

# 1. Introduction

The biomedical applications of superparamagnetic iron oxide nanoparticles (SPIONs) in magnetic resonance imaging, targeted therapy and cell labelling have been extensively studied. Although many applications in the diagnosis and treatment of diseases have shown good potential, there are still some controversial results concerning the cytotoxic effects from the use of SPIONs [1,2]. Some studies have reported that SPIONs are biologically benign [3], whereas other researchers believe that SPIONs have potential toxicity including organ toxicity and genotoxicity [1].

Magnetic $Fe_3O_4$ nanoparticles ($Fe_3O_4$-NPs) are a new type of nanomaterial with large specific surface area, high biocompatibility and biodegradability, which fall under the SPIONs category and have great potential for development for use in biomedicine [4]. $Fe_3O_4$-NPs have unique physical and chemical properties, including the following: superparamagnetism, magnetocaloric effects and peroxidase-like activity. The superparamagnetism of $Fe_3O_4$-NPs can be targeted to regulate $Fe_3O_4$-NPs *in vivo* through an external magnetic field for use in the targeted transport of drugs [5,6]. The magnetocaloric effect of $Fe_3O_4$-NPs can be used to convert their electromagnetic energy into heat energy through a repeated magnetization process, which can be used for tumour hyperthermia [7]. The peroxidase activity of $Fe_3O_4$-NPs can catalyse the degradation of $H_2O_2$ in acidic or neutral pH environments [8]. In recent years, $Fe_3O_4$-NPs have also shown great value by inhibiting bacterial activity [9–12]. Shi *et al.* [11] reported that the $Fe_3O_4$-NPs may be a potential antibiotic alternative to control *Salmonella enteritidis* infection during clinical therapy and in poultry industry operations. Based on the potential function of $Fe_3O_4$-NPs in disease diagnosis and treatment, it is necessary to further study the cytotoxic effects of $Fe_3O_4$-NPs before they are applied in various fields.

Although some reports on the cytotoxic effects of $Fe_3O_4$-NPs have been reported, most of them focus on the research in mammals (rats, mice or humans) [13–16]. There are few reports on the application of $Fe_3O_4$-NPs in animal husbandry [17,18]. There are zoonotic infectious diseases caused by bacteria such as *Salmonella enteritidis*, which has led to large numbers of deaths in humans and caused economic losses in animal husbandry. Our previous research found that $Fe_3O_4$-NPs could effectively control *S. enteritidis* infection in chicken LMH cells [11]. As macrophages play an important role in natural immunity and acquired immunity during *S. enteritidis* infection, we choose the macrophages of poultry as the research object.

Chicken HD11 macrophages are a kind of immortalized cell line formed by transforming chicken bone marrow cells through the replication-deficient avian leukaemia virus MC29 strain, which has been widely used in the study of infection immunity and its mechanism in bacteria or viruses [19]. HD11 cells share many similarities with normal chicken macrophages and have the advantage of rapid subculture growth; hence, they can be used as an ideal model of chicken macrophages for *in vitro* studies [20–22]. Moreover, poultry belongs to Aves and has many different characteristics in structure and function from mammals. Is there any difference between the application of $Fe_3O_4$-NPs in mammals and poultry? No research has reported such results yet. Therefore, we studied the cytotoxic effects of $Fe_3O_4$-NPs of different sizes on chicken macrophages in terms of their oxidative effects, apoptosis and pro-inflammatory cytokine secretion, laying a foundation for the potential application of $Fe_3O_4$-NPs as potential antibiotic substitutes in the poultry industry.

# 2. Material and methods

## 2.1. Characterization of $Fe_3O_4$ nanoparticles

$Fe_3O_4$-NPs with diameters of approximately 60, 120 and 250 nm were prepared by a hydrothermal method with $FeCl_3$ and $NaAc·3H_2O$ as raw materials. A 0.4 or 0.6 g sample of $FeCl_3$ and 3.6 g of $NaAc·3H_2O$ were added into the mixed solvent of 10 ml of glycol and 30 ml of diethylene glycol and stirred until fully dissolved by ultrasound, and these solutions were used to prepare 60 or 120 nm $Fe_3O_4$-NPs. A 0.82 g sample of $FeCl_3$ and 3.6 g of $NaAc·3H_2O$ were added into 40 ml of glycol and stirred to achieve full dissolution by ultrasound, which was used to prepare the 250 nm $Fe_3O_4$-NPs. After complete dissolution, the mixture was transferred into a 50 ml Teflon-sealed autoclave and heated at 200°C for 12 h. After being naturally cooled to room temperature, the reactants were washed three times with water and three times with ethanol and then dried at 60°C for 6 h under a vacuum. The morphology, particle size and size distribution of the $Fe_3O_4$-NPs were measured using scanning electron microscopy (SEM) (S-4800, Japan) and transmission electron microscopy (TEM) (Tecnai G2

F30 S-TWIN, America). The diameters of the $Fe_3O_4$-NPs in the dispersion were determined using the dynamic light scattering (DLS) technique (Nano ZS90, England). The zeta potential of the $Fe_3O_4$-NPs was also measured using a laser particle size and zeta potential analyser (Nano ZS90, England).

## 2.2. Cell cultures

The chicken macrophage HD11 cells were gifted from Professor Jiao of Yangzhou University. The HD11 cells were cultured in 1640 medium (HyClone, Utah, USA) containing 10% foetal bovine serum (FBS) (Gibco, CA, USA) and 1% penicillin and streptomycin (Solarbio, Beijing, China) at 37°C and 5% $CO_2$.

## 2.3. Cell viability assay

Cell viability was measured by the CCK-8 assay kit (Dojindo, Japan) according to the manufacturer's instructions. Briefly, HD11 cells were seeded at approximately $2\times10^5$ cells $ml^{-1}$ in 96-well plates and treated with 60, 120 and 250 nm $Fe_3O_4$-NPs for 24 h at concentrations of 50, 100, 200 and 400 µg $ml^{-1}$. A 10 µl aliquot of CCK-8 reagent was added to each well, and the cells were incubated for another 2 h. The absorbance at 450 nm was measured with a microplate reader (Infinite M200 Pro, Tecan, Switzerland) after incubation.

## 2.4. Determination of the oxidative index malondialdehyde, superoxide dismutase and reactive oxygen species

HD11 cells in 24-well plates were incubated with different sizes and different concentrations of $Fe_3O_4$-NPs in 1640 medium with 10% FBS. After 24 h of exposure, the cell culture supernatant was collected and centrifuged at 1000 r.p.m. for 10 min in preparation to measure the malondialdehyde (MDA) and superoxide dismutase (SOD). The levels of MDA and SOD were measured by ELISA methods (Nanjing Jiancheng Bioengineering Institute, Nanjing, China). The absorbance of MDA/SOD was measured at 532 nm/450 nm with a microplate reader (Infinite M200 Pro, Tecan, Switzerland).

Reactive oxygen species (ROS) levels were determined by a reactive oxygen species assay kit (Beyotime Biotechnology Ltd., Shanghai, China). HD11 cells in 96-well plates were incubated with $Fe_3O_4$-NPs of different sizes and at different concentrations in 1640 medium with 10% FBS. After 24 h of exposure to the $Fe_3O_4$-NPs, the cells were incubated with 10 µM DCFH-DA in medium without FBS at 37°C and 5% $CO_2$ for 30 min. Then, DCFH-DA was removed, and the cells were washed twice with PBS. The DCF fluorescence was monitored at 485 nm excitation and 520 nm emission with a microplate reader (Infinite M200 Pro, Tecan, Switzerland).

## 2.5. Determination of the extent of apoptosis

Apoptosis of the HD11 cells exposed to $Fe_3O_4$-NPs of various sizes and concentrations was measured using an Annexin V-FITC apoptosis analysis kit (BD, NJ, USA) according to the manufacturer's instructions. HD11 cells in 12-well plates were incubated with $Fe_3O_4$-NPs of different sizes at different concentrations in 1640 medium with 10% FBS. After 24 h of exposure, the cells were collected and washed twice with cold PBS while being stirred at 1000 r.p.m. for 5 min. Then, the cells were adjusted to a concentration of $1 \times 10^6$ cells $ml^{-1}$ with 1 × binding buffer. A total of 100 µl of cell suspension was placed in a 5 ml Falcon tube, and 5 µl of FITC Annexin V was added to each tube in the dark for 20 min at room temperature. Then, 5 µl of PI was added to each tube in the dark for 5 min, and 400 µl of 1 × binding buffer was added to each tube. All samples were analysed using flow cytometry (BD LSRFortessa™, USA) within 1 h.

## 2.6. Determination of interleukin-1$\beta$ expression

HD11 cells in 12-well plates were incubated with $Fe_3O_4$-NPs of different sizes in 1640 medium with 10% FBS. After 24 h, the cells were collected using Trizol (Invitrogen, Carlsbad, CA) to extract the total RNA according to the manufacturer's protocol. RT-PCR was performed to quantitate interleukin-1$\beta$ (IL-1$\beta$) mRNA using StepOnePlus™ (Applied Biosystems by Life Technologies, USA). The primers of IL-1$\beta$ were designed and synthesized by the Invitrogen Biotechnology Co., Ltd (Shanghai, China; table 1).

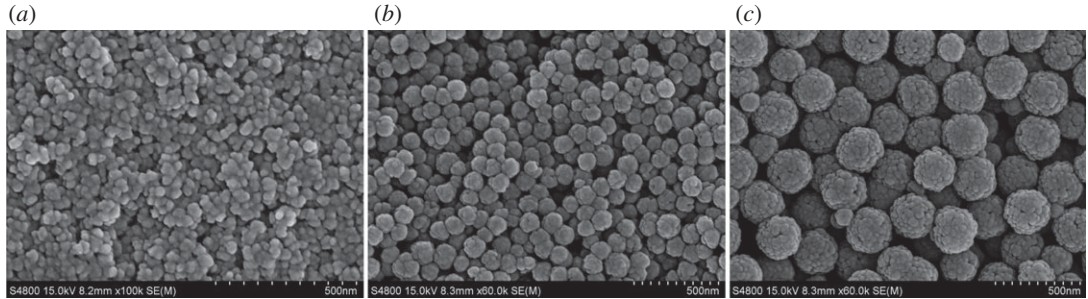

**Figure 1.** SEM images of Fe$_3$O$_4$-NPs. (*a*) 60 nm; (*b*) 120 nm; (*c*) 250 nm.

**Table 1.** Primer sequences for the target genes.

| gene | primer position | primer sequence (5′→3′) | product size (bp) |
|---|---|---|---|
| B-actin | forward | ACACCCACACCCCTGTGATGAA | 136 |
| | reverse | TGCTGCTGACACCTTCACCATTC | |
| IL-1$\beta$ | forward | TGGGCATCAAGGGCTACA | 244 |
| | reverse | TCGGGTTGGTTGGTGATG | |

## 2.7. Statistical analysis

All results are presented as the mean ± standard error (s.e.). One-way analysis of variance (ANOVA) (multiple comparisons using LSD test) was used to evaluate the multiple comparisons among different sizes of Fe$_3$O$_4$-NPs groups. $p < 0.05$ indicated a significant difference.

# 3. Results and discussion

## 3.1. Synthesis and characterization of Fe$_3$O$_4$ nanoparticles

Micrographs obtained by SEM (figure 1) and TEM (figure 2) showed that the nanoparticles were spherical and uniform in shape. The hydrodynamic diameters of the Fe$_3$O$_4$-NPs were measured by DLS. As shown in figure 3, the average hydrodynamic diameters of the 60, 120 and 250 nm Fe$_3$O$_4$-NPs were 68 ± 2.40, 121.3 ± 3.04 and 250 ± 6.79 nm, respectively, which was in good agreement with the TEM results; the zeta potentials were +19.4 ± 1.52, +18.9 ± 1.48 and +20.3 ± 1.6 mV, respectively, indicating that the nanoparticles were positively charged, which is beneficial for endocytosis.

## 3.2. Cytotoxic effects of Fe$_3$O$_4$ nanoparticles of different sizes

A CCK-8 assay was performed to evaluate the effect of different sized Fe$_3$O$_4$-NPs on HD11 cell viability. The HD11 cells were incubated with various concentrations of 60, 120 and 250 nm Fe$_3$O$_4$-NPs. As shown in figure 4, the cell viability of the 250 nm Fe$_3$O$_4$-NPs group was significantly greater than those of the 60 nm and 120 nm groups at the concentration of 50 µg ml$^{-1}$, and the activity levels of the 120 nm and 250 nm groups were significantly greater than that of the 60 nm group at a concentration of 100 µg ml$^{-1}$. These data suggested that the 60 nm Fe$_3$O$_4$-NPs produced a stronger cytotoxic effect when added at the same concentration as the other groups.

Many reports indicate that iron oxide nanoparticles are biologically safe because of their biocompatibility and high tolerance by cells [3,23,24]. However, other reports indicated that these nanoparticles had the potential to produce toxic effects in cells, and their toxic effects were related to their size, concentration, time, shape and the cell type [13,15,25,26]. The results from the present study suggest that the extent of Fe$_3$O$_4$-NPs toxic effects in HD11 cells is affected by the size of the Fe$_3$O$_4$-NPs. In our study, the 60 nm Fe$_3$O$_4$-NPs produced stronger cytotoxicity at the same concentration as the other groups according to the CCK-8 assay. By contrast, the 250 nm Fe$_3$O$_4$-NPs showed few cytotoxic effects in HD11 cells. These data suggested that the cytotoxic effects of small-sized Fe$_3$O$_4$-NPs were greater than those of large-sized Fe$_3$O$_4$-NPs for HD11 cells. Consistent with previous

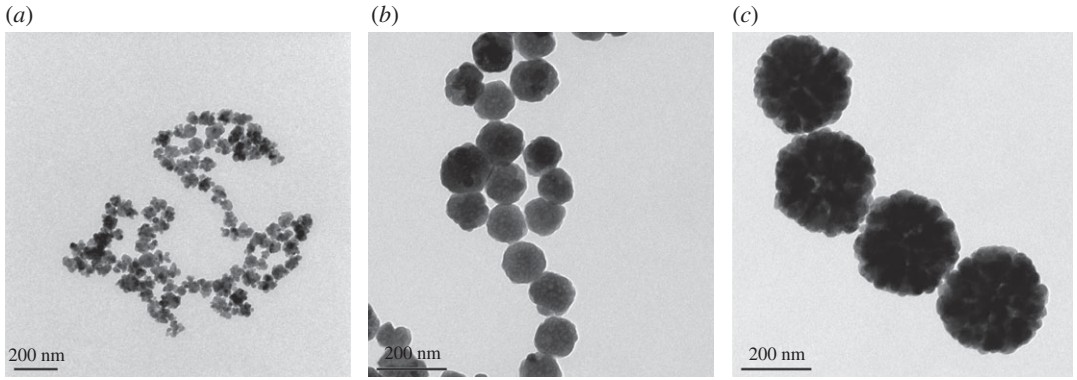

**Figure 2.** TEM images of Fe$_3$O$_4$-NPs. (*a*) 60 nm; (*b*) 120 nm; (*c*) 250 nm.

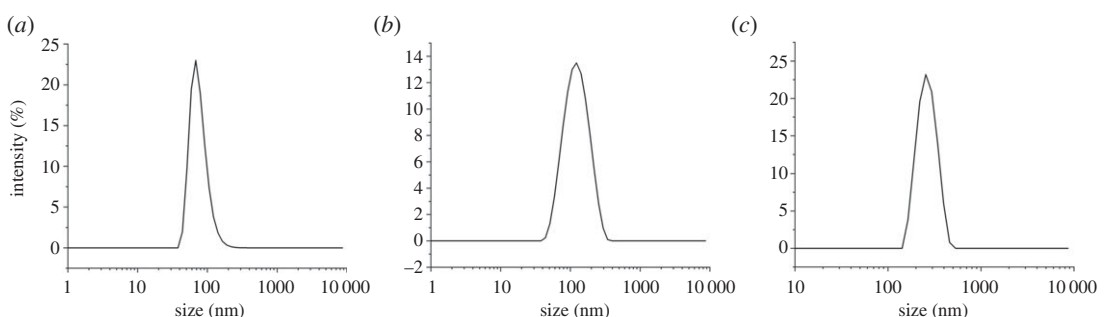

**Figure 3.** Zeta potential of Fe$_3$O$_4$-NPs. (*a*) 60 nm; (*b*) 120 nm; (*c*) 250 nm.

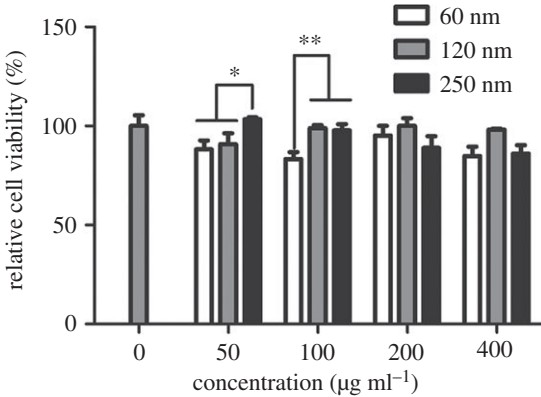

**Figure 4.** Effects of 60, 120 and 250 nm Fe$_3$O$_4$-NPs on the viability of HD11 cells. Control cells cultured in nanoparticle-free 1640 medium were processed in parallel to the treatment groups. The results of the CCK-8 assay were expressed as a percentage of the control. The data are expressed as the mean ± s.e., $^{**}p < 0.01$, $^{*}p < 0.05$. Fe$_3$O$_4$-NPs, magnetic Fe$_3$O$_4$ nanoparticles.

reports, Chen *et al.* [16] reported that smaller SPIONs produced stronger cytotoxicity in mouse bone marrow-derived macrophages. However, Gong *et al.* [25] evaluated 30 and 50 nm GoldMag nanoparticles (GMNPs) with SPIO as the core and gold coating, and they found that the nanotoxicity produced by the 50 nm GMNPs was significantly higher than that produced by 30 nm GMNPs on human umbilical vein endothelial cells when administered at identical levels of concentration and exposure. It may be that the size of iron nanoparticles compared in each experiment was different, and the different cells may have different responses to iron nanoparticles. In addition, the toxic effect of iron nanoparticles in cells is also related to the material with which they are coated. Therefore, it is important to fully investigate the biosafety of nanoparticles before they are used in biomedical or livestock production.

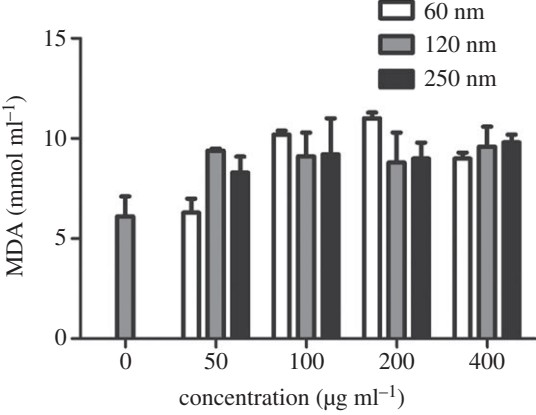

**Figure 5.** Effects of 60, 120 and 250 nm $Fe_3O_4$-NPs on MDA production in HD11 cells. Control cells cultured in nanoparticle-free 1640 medium were processed in parallel to the treatment groups. The data are expressed as the mean $\pm$ s.e. $Fe_3O_4$-NPs, magnetic $Fe_3O_4$ nanoparticles.

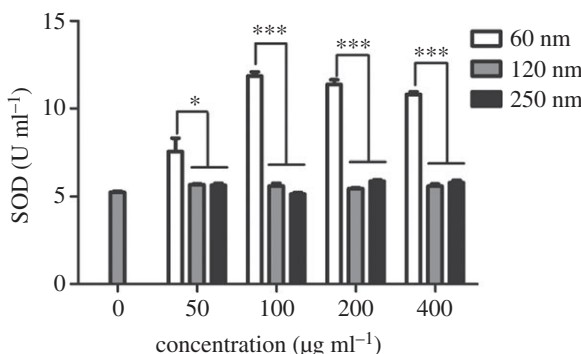

**Figure 6.** Effects of 60, 120 and 250 nm $Fe_3O_4$-NPs on SOD activities in HD11 cells. Control cells cultured in nanoparticle-free 1640 medium were processed in parallel to the treatment groups. The data are expressed as the mean $\pm$ s.e., $^{***}p < 0.001$, $^{*}p < 0.05$. $Fe_3O_4$-NPs, magnetic $Fe_3O_4$ nanoparticles.

## 3.3. Effects of $Fe_3O_4$ nanoparticles on malondialdehyde, superoxide dismutase and reactive oxygen species levels

As shown in figure 5, no significant difference in the MDA level was observed among the three sizes of $Fe_3O_4$-NPs. However, the activity level of SOD differed, as presented in figure 6. HD11 cells showed significant increases in SOD activity when exposed to 60 nm $Fe_3O_4$-NPs. SOD levels of the 60 nm group were significantly higher than those of the 120 and 250 nm groups at 50 $\mu$g ml$^{-1}$. SOD levels of the 60 nm group were significantly higher, showing extreme differences, than those of the 120 and 250 nm groups at 100, 200 and 400 $\mu$g ml$^{-1}$. The results indicated that the 60 nm group had a significant effect on SOD levels, possibly due to the self-regulation of cells. Figure 7 showed the effects of different sizes of $Fe_3O_4$-NPs on ROS generation in HD11 cells. At concentrations of 50, 100 and 200 $\mu$g ml$^{-1}$, ROS levels of the 120 and 250 nm groups were significantly lower, showing extreme differences, than those of the 60 nm group. The ROS level of the 250 nm group was significantly lower than that of the 60 nm group at 400 $\mu$g ml$^{-1}$. These data suggested that the 60 nm $Fe_3O_4$-NPs produced more ROS when added at the same concentration as the other groups.

It has been reported that iron oxide nanoparticles induce cytotoxicity by activating the oxidative stress response [27,28]. The imbalance between oxidants and antioxidants favours the oxidants and potentially leads to damage known as 'oxidative stress' [29]. MDA is the final decomposition product of membrane lipid peroxidation and widely used as a marker of oxidative lipid injury. SOD is a natural scavenger of oxygen-free radicals in organisms, which can eliminate ROS and inhibit the harmful effects of oxidant molecules on tissues and cells. ROS produced in biological systems play an important role in cell damage, cell apoptosis and other metabolic activities. Sarkar & Sil [28] found that iron oxide nanoparticles increased MDA and ROS levels in murine hepatocytes and decreased the activity of the antioxidant enzyme SOD, which correspondingly caused oxidative stress. In our study,

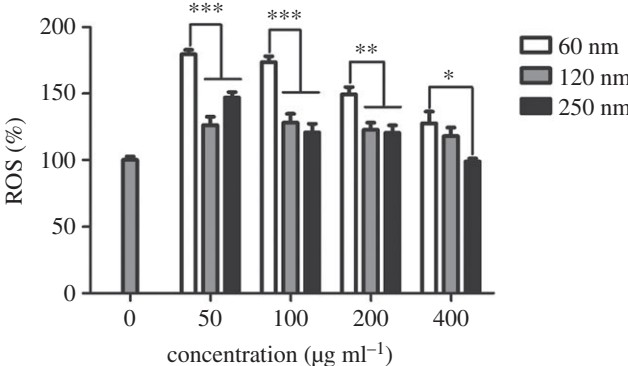

**Figure 7.** Effects of 60, 120 and 250 nm $Fe_3O_4$-NPs on ROS generation in HD11 cells. Control cells cultured in nanoparticle-free 1640 medium were processed in parallel to the treatment groups. The data are expressed as the mean ± s.e., ***$p < 0.001$, **$p < 0.01$, *$p < 0.05$. $Fe_3O_4$-NPs, magnetic $Fe_3O_4$ nanoparticles.

we focused on the effects of different sizes of $Fe_3O_4$-NPs on cell oxidation. Compared with the control (nanoparticle-free) group, the three sizes of $Fe_3O_4$-NPs, 60, 120 and 250 nm, showed a trend of increasing MDA, but the MDA levels of the three sizes were not significantly different in our study. Among the three sizes, the SOD level of the 60 nm $Fe_3O_4$-NPs was the highest, whereas it was not enough to eliminate the production of ROS associated with $Fe_3O_4$-NPs of the same size because the ROS level of the group with $Fe_3O_4$-NPs of 60 nm was also the highest among the ROS levels measured in this study. The SOD levels for the $Fe_3O_4$-NPs of 120 and 250 nm were relatively low, possibly due to the compensatory reaction of the antioxidant system cells used to eliminate ROS production.

## 3.4. Effect of $Fe_3O_4$ nanoparticles on the extent of apoptosis

Figure 8 shows the effects of different sizes of $Fe_3O_4$-NPs on the extent of apoptosis of HD11 cells. At concentrations of 50 and 100 $\mu g\,ml^{-1}$, the extent of apoptosis of the 250 nm group was significantly less than those of the 60 and 120 nm groups. At a concentration of 200 $\mu g\,ml^{-1}$, the extent of apoptosis of the 120 nm group was significantly less than that of the 60 nm group, and the extent of apoptosis of the 250 nm group was extremely significantly less than that of the 60 nm group. Additionally, the extent of apoptosis of the 250 nm group was significantly lower than that of the 60 nm group at 400 $\mu g\,ml^{-1}$. These data suggested that the extent of apoptosis was in the order 60 nm > 120 nm > 250 nm.

We observed that the greatest extent of apoptosis was caused by the 60 nm $Fe_3O_4$-NPs in this study. This finding was consistent with previous reports suggesting that the high levels of ROS could cause apoptosis or cell death [30,31]. When the production of intracellular ROS exceeds the threshold of cell ability for antioxidant defences, macromolecules such as DNA, proteins and lipids are damaged, which could lead to this pathophysiological state [32]. Therefore, the results of our study showed that the apoptosis induced by $Fe_3O_4$-NPs was mainly related to oxidative stress due to increased ROS production, and the apoptosis induced by small-sized $Fe_3O_4$-NPs was relatively more serious. However, there may be other pathways involved in apoptosis such that further study is needed.

## 3.5. Effect of $Fe_3O_4$ nanoparticles on IL-1$\beta$ expression

In addition, we suspect that the cytotoxicity of $Fe_3O_4$-NPs may be related to the secretion of inflammatory cytokines in macrophages. Based on the above results, the cell activity of the 60 nm group was significantly lower than those of the 120 and 250 nm groups at the concentration of 100 $\mu g\,ml^{-1}$, and the biological safety of 250 nm $Fe_3O_4$-NPs was better. Therefore, we chose 60 and 120 nm $Fe_3O_4$-NPs at 100 $\mu g\,ml^{-1}$ to detect their effects on inflammatory cytokines in macrophages. As shown in figure 9, the IL-1$\beta$ secreted by macrophages was increased by 60 and 120 nm $Fe_3O_4$-NPs, and the IL-1$\beta$ expression of the 60 nm group was the highest, which was consistent with the maximum cytotoxicity.

Macrophages are one of the main cells involved in the inflammatory response. IL-1$\beta$ is one of the important inflammatory cytokines, which mediates the important process of inflammation [33]. Previous studies have reported that iron oxide nanoparticles could induce the cellular inflammatory response and

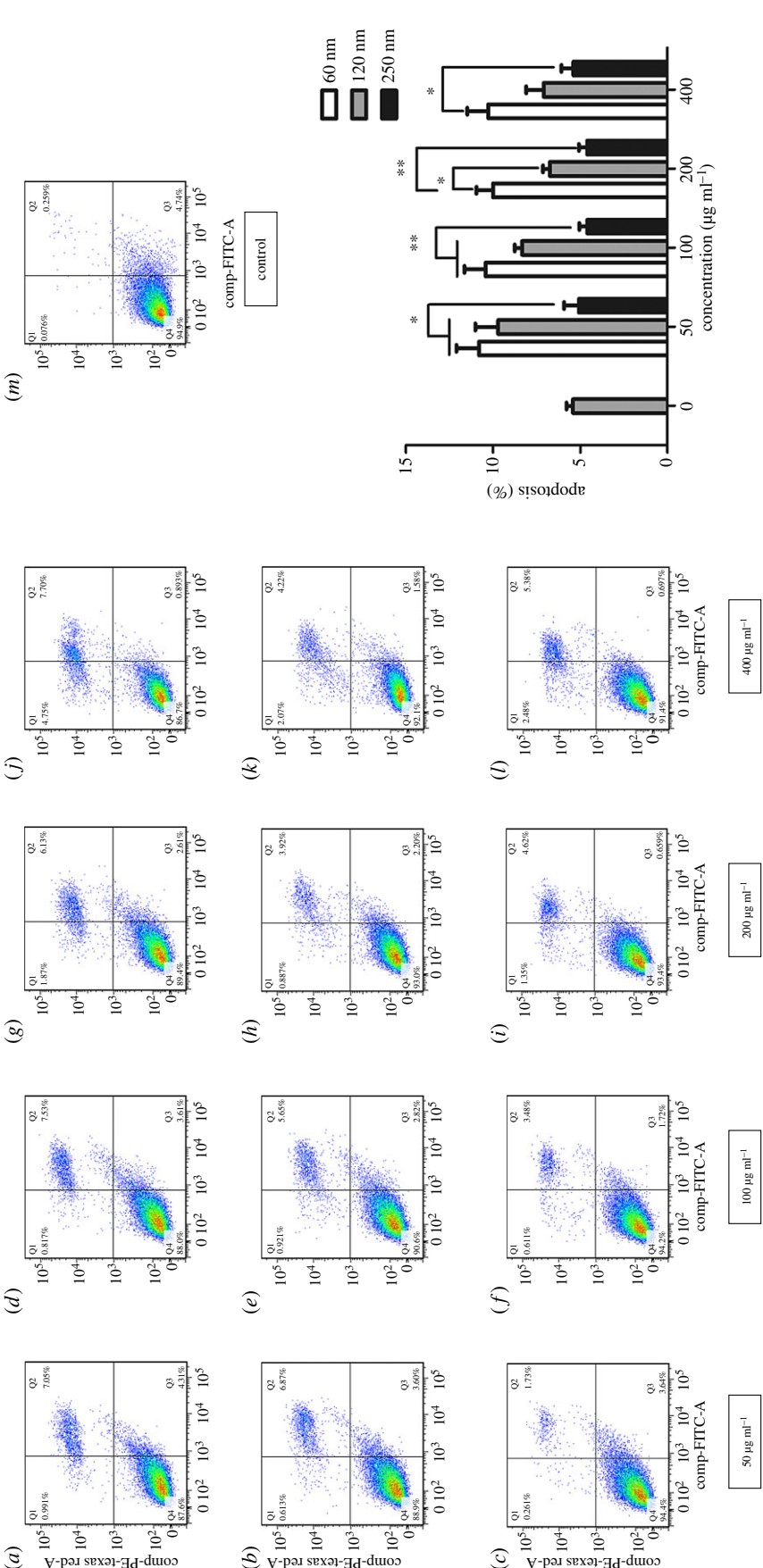

**Figure 8.** Effects of 60, 120 and 250 nm $Fe_3O_4$-NPs on the extent of apoptosis in HD11 cells. $(a–c)$ 50 µg ml$^{-1}$ of 60, 120 and 250 nm $Fe_3O_4$-NPs; $(d–f)$ 100 µg ml$^{-1}$ of 60, 120 and 250 nm $Fe_3O_4$-NPs; $(g–l)$ 200 µg ml$^{-1}$ of 60, 120 and 250 nm $Fe_3O_4$-NPs; $(j–l)$ 400 µg ml$^{-1}$ of 60, 120 and 250 nm $Fe_3O_4$-NPs; $(m)$ the control group without nanoparticle. The data are expressed as the mean ± s.e., $**p < 0.01$, $*p < 0.05$. $Fe_3O_4$-NPs, magnetic $Fe_3O_4$ nanopartides.

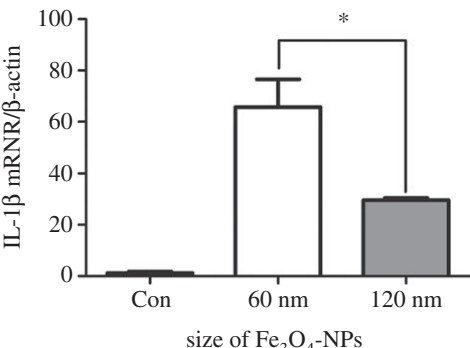

**Figure 9.** Effects of 60 and 120 nm $Fe_3O_4$-NPs at the concentration of 100 μg ml$^{-1}$ on IL-1β expression of HD11 cells. Control cells cultured in nanoparticle-free 1640 medium were processed in parallel to the treatment groups. The data are expressed as the mean ± s.e., *$p < 0.05$. $Fe_3O_4$-NPs, magnetic $Fe_3O_4$ nanoparticles.

increase the secretion of pro-inflammatory cytokines in human cells or mouse cell lines [34–36]. However, the overproduction of inflammatory cytokines could cause many inflammatory diseases, such as lung injury, obesity and diabetes [37,38]. It has also been reported that iron oxide nanoparticles could inhibit tumour growth by pro-inflammatory immune responses [39]. In general, the secretion of inflammatory cytokines by macrophages is a double-edged sword. On the one hand, it can cause cell damage, degeneration or death; on the other hand, it can enhance cell resistance to disease and make the intracellular and extracellular environment reach a new balance. In this study, we found that $Fe_3O_4$-NPs could increase the expression of IL-1β in HD11 cells, and the IL-1β expression was the highest when induced by 60 nm $Fe_3O_4$-NPs, which was consistent with the maximum cytotoxicity induced by 60 nm $Fe_3O_4$-NPs. This suggested that the inflammatory response may also be one of the important mechanisms for the cytotoxicity induced by $Fe_3O_4$-NPs.

# 4. Conclusion

In conclusion, the smaller $Fe_3O_4$-NPs produced stronger cytotoxicity in chicken macrophage cells, which may be related to the oxidative stress and apoptosis induced by the increased ROS production as well as the increased expression of pro-inflammatory cytokines. Based on this study, 250 nm $Fe_3O_4$-NPs have good biological safety, which lays a foundation for the potential application of $Fe_3O_4$-NPs in the poultry industry. The study of $Fe_3O_4$-NPs as potential antibiotic substitutes and their mechanism requires further experiments.

Data accessibility. All the data are provided in the electronic supplementary material.

Authors' contributions. S.Z. carried out the cell laboratory work, participated in the design of the study, analysis, interpreted data and drafted the manuscript; S.W. carried out the work of preparing $Fe_3O_4$-NPs and analysing their characterization; Y.S. carried out the work of detecting the oxidative index and apoptosis level; M.X. carried out the work of editing the version of the manuscript; Y.X. carried out the work of detecting the pro-inflammatory cytokine secretion level and analysing data; L.G. and S.S conceived the study, designed the study and critically revised the manuscript. All authors gave final approval for publication and agree to be held accountable for the work performed therein.

Competing interests. We declare that we have no conflicts of interest.

Funding. This work was financially supported by the National Natural Science Foundation of China (31702132), the Innovation Capacity Building Program of Jiangsu Province (BM2018026) and the Jiangsu Provincial Key Laboratory of Poultry Genetics & Breeding (JQLAB-ZZ-201804).

Acknowledgement. The paper was made better thanks to useful comments and modifications from Ming Xu.

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
