## [Reviewer comments · Royal Society Open Science]

Review History

RSOS-191561.R0 (Original submission)

Review form: Reviewer 1

Is the manuscript scientifically sound in its present form?

No

Are the interpretations and conclusions justified by the results?

No

Is the language acceptable?

Yes

Do you have any ethical concerns with this paper?

No

Have you any concerns about statistical analyses in this paper?

No

Recommendation?

Reject

Comments to the Author(s)

The manuscript describes the toxicity potential of different sized Fe₃O₄ nanoparticles (NPs) in chicken macrophage cells (HD11 cells).

I have some serious concerns and doubt on the results and interpretation especially in the biology part. Besides scientific and technical issues, it seems that the manuscript appears like a draft and it is not suitable for publication in "Royal Society Open Science". Therefore, it should be rejected.

Some specific concerns:

i) Explanation of all the data is poor.

ii) '...there are still some controversial results concerning the cytotoxic effects from use of SPIONs...'

The controversial results should be described in Introduction section.

iii) The synthesis procedure did not clarify the conditions of formation of three different sized (60 nm, 120 nm, 250 nm) nanoparticles.

iv) At which wavelength the absorbance of MDA/SOD was measured using ELISA? It should be mentioned in experimental section.

v) Fluorescence microscopy or confocal microscopy would be more convincing for determination of ROS. The present work lacks of state-of-the art techniques.

vi) Why is there no much difference in size of NPs obtained from TEM and DLS, especially for 120 nm and 250 nm NPs? Generally, the hydrodynamic size of NPs obtained in DLS is much greater than the size of NPs obtained in TEM.

vii) The interpretation of cell viability data (Figure 4) and the execution of the experiment are faulty. Although 60 nm NPs exerted slight cytotoxicity in HD11 cells at 50-100 µg/mL as compared to control untreated cells, it did not exert any cytotoxicity (almost nil) even at higher dose 200 µg/mL. Again at 400 µg/mL dose, it exhibited similar slight toxicity as observed for 50 µg/mL dose. So the result is not consistent as per the dosing.

It seems that all the NPs (60 nm, 120 nm and 250 nm) are showing very slight toxicity, as compared to untreated control cells at a broad concentration range (50-400 µg/mL). However, there is no dose dependent consistent result for 60 nm NPs, from which it can be interpreted that 60 nm NPs are more toxic than others. e.g. at 200 µg/mL dose, 250 nm NPs are more toxic than 60 nm NPs and at 400 µg/mL dose, the toxicity for 60 nm and 250 nm NPs are similar. Therefore, the interpretations of further study results which are based on the cell viability data are also misleading.

viii) The cytotoxic potential of 60 nm NPs at 200 µg/mL is lower than that of 50 and 100 µg/mL dose. Why did it happen?

ix) Why did you check MDA, SOD and ROS levels and what is the outcome of these data with respect to the cytotoxicity potential of NPs? It should be clearly explained in corresponding result sections.

x) How did the self-regulation of cells influence the higher SOD level for the treatment with 60 nm NPs, unlike other NPs?

xi) Why is the ROS level gradually reduced with increasing concentration of 60 nm NPs? The result is absurd.

xii) The text within Figure 8 is not visible. It should be presented in a proper way. Apoptosis cell % is not visible in the diagram and histogram.

xiii) Apoptosis data does not corroborate with the cell viability data. Cell viability shows almost biocompatible nature of 120 nm NPs except at 50 µg/mL dose. But apoptosis study exhibited higher % apoptosis for 120 nm NPs treatment at all concentrations as compared to control untreated cells.

Why is there reduction in % apoptosis with increasing concentration of 120 nm NPs?

xiv) The English should be more polished.

xv) Conclusion section is poorly written.

Review form: Reviewer 2

Is the manuscript scientifically sound in its present form?

Yes

Are the interpretations and conclusions justified by the results?

Yes

Is the language acceptable?

Yes

Do you have any ethical concerns with this paper?

Yes

Have you any concerns about statistical analyses in this paper?

No

Recommendation?

Major revision is needed (please make suggestions in comments)

Comments to the Author(s)

The manuscripts reveals an interesting study of cytotoxicity of Fe₃O₄ nano particles on chicken macrophage cells to assess its effectiveness against *S. Entreditis* infection in poultry.

The report reveals that cytotoxicity of 250 nm particles is least and the particle of this size may be used as an antibiotic alternative. However, I have some concerns if such big size nanoparticles can be used in drug delivery and authors must add some literature reports wherein particles greater than 200 nm has been used. Authors should also convince readers about the effectiveness of the 250 nm Fe₃O₄ nanoparticles to cross blood brain barrier since mostly for the drug delivery nanoparticle size used is between 10-100 nm.

Review form: Reviewer 3

Is the manuscript scientifically sound in its present form?

Yes

Are the interpretations and conclusions justified by the results?

Yes

Is the language acceptable?

Yes

Do you have any ethical concerns with this paper?

No

Have you any concerns about statistical analyses in this paper?

No

Recommendation?

Major revision is needed (please make suggestions in comments)

Comments to the Author(s)

Manuscript ID: RSOS-191561

Title: Cytotoxicity studies of Fe₃O₄ nanoparticles in chicken macrophage cells

Authors: Zhang, S., et al.

This manuscript describes the preparation of Magnetic Fe₃O₄ nanoparticles and examines them for their activity (oxidative index (malondialdehyde, superoxide dismutase and reactive oxygen species formation), apoptosis and pro-inflammatory cytokine secretion level in chicken macrophage cells as a function of nanoparticle size. They find that, when there are differences, there is greater activity in smaller nanoparticles.

The authors describe research that utilized magnetic nanoparticles of 60, 120 and 250 nm diameters. They provide a general approach for how the nanoparticles were prepared but not how the approach was used to get just 60 or 120 or 250 nm particles. This point is fairly important in the event that someone wants to reproduce the authors work.

The authors note that toxicity of nanoparticles should be examined prior to their use in vivo (e.g., p 7, lines 29-31, 'Therefore, it is important to fully investigate the biosafety of nanoparticles before they are used in biomedical or livestock production.' What is the fate of the nanoparticles? This is relevant here because if they are retained and the livestock eaten, they may also affect humans and so toxic effects in humans also needs to be examined.

The expected use of these iron nanoparticles should be stated. The authors conclude the manuscript with the statement 'Based on this study, 250 nm Fe₃O₄-NPs have good biological safety and maybe be used in disease diagnosis and substitution research in poultry industry.' (p. 9, conclusion section, lines 39-41). Are the authors suggesting the nanoparticles are going to be used to treat infections and take advantage of the ROS generation or, will the particles simply act as carriers. In the latter case, the particles studied in this manuscript would not be relevant as there surfaces were unmodified. The authors, at a minimum, need to describe how the nanoparticles are expected to be used. They may also want to state that in vivo degradation of particles with coatings are bearing 'payloads' (e.g., drugs) may result in the bare magnetic nanoparticles.

Misc:

Abstract: The sentence 'The results revealed that the cell viability of the 250 nm Fe₃O₄-NPs group was significantly greater ($P < 0.05$) than that of the 60 nm and 120 nm groups at a concentration of 50 μ g/mL, and the activity level of the 120 nm and 250 nm groups was significantly greater ($P < 0.01$) than that of the 60 nm group at a concentration of 100 μ g/mL.' is poorly written/confusing. If I understand it correctly, the authors are simply providing a rank

order and could say that the smaller the particle size the greater the activity. If they feel it necessary, the qualifiers regarding concentration, etc. can be added.

The authors make some statements for which citations should be added

p. 3, (Introduction section), lines 41-43, it is stated that 'There are few reports on the application of Fe₃O₄-NPs in animal husbandry.' If there are any reports, a representative citation(s) should be provided.

p. 3, lines 45-47 reads 'Our previous research found that Fe₃O₄-NPs could effectively control *S. Enteritidis* infection in chicken LMH cells.' This statement should be cited. If not published, then it should say 'unpublished research.'

p. 4, (under Materials and methods) line 21 reads 'fully dissolving in glycol...' should read fully dissolved in glycol...' Also, is this ethylene glycol (vs propylene glycol, etc.)?

Decision letter (RSOS-191561.R0)

30-Jan-2020

Dear Miss Zhang:

Title: Cytotoxicity studies of Fe₃O₄ nanoparticles in chicken macrophage cells
Manuscript ID: RSOS-191561

The editor assigned to your manuscript has now received comments from reviewers. We would like you to revise your paper in accordance with the referee and Subject Editor suggestions which can be found below (not including confidential reports to the Editor). Please note this decision does not guarantee eventual acceptance.

Please submit your revised paper before 22-Feb-2020. Please note that the revision deadline will expire at 00.00am on this date. If we do not hear from you within this time then it will be assumed that the paper has been withdrawn. In exceptional circumstances, extensions may be possible if agreed with the Editorial Office in advance. We do not allow multiple rounds of revision so we urge you to make every effort to fully address all of the comments at this stage. If deemed necessary by the Editors, your manuscript will be sent back to one or more of the original reviewers for assessment. If the original reviewers are not available we may invite new reviewers.

RSC Associate Editor:
Comments to the Author:
(There are no comments.)

RSC Subject Editor:
Comments to the Author:
(There are no comments.)

Reviewers' Comments to Author:
Reviewer: 1

Comments to the Author(s)
The manuscript describes the toxicity potential of different sized Fe₃O₄ nanoparticles (NPs) in chicken macrophage cells (HD11 cells).

I have some serious concerns and doubt on the results and interpretation especially in the biology part. Besides scientific and technical issues, it seems that the manuscript appears like a draft and it is not suitable for publication in "Royal Society Open Science". Therefore, it should be rejected.

Some specific concerns:

- i) Explanation of all the data is poor.
- ii) '...there are still some controversial results concerning the cytotoxic effects from use of SPIONs...'
The controversial results should be described in Introduction section.
- iii) The synthesis procedure did not clarify the conditions of formation of three different sized (60 nm, 120 nm, 250 nm) nanoparticles.
- iv) At which wavelength the absorbance of MDA/SOD was measured using ELISA? It should be mentioned in experimental section.

v) Fluorescence microscopy or confocal microscopy would be more convincing for determination of ROS. The present work lacks of state-of-the art techniques.

vi) Why is there no much difference in size of NPs obtained from TEM and DLS, especially for 120 nm and 250 nm NPs? Generally, the hydrodynamic size of NPs obtained in DLS is much greater than the size of NPs obtained in TEM.

vii) The interpretation of cell viability data (Figure 4) and the execution of the experiment are faulty. Although 60 nm NPs exerted slight cytotoxicity in HD11 cells at 50-100 $\mu\text{g}/\text{mL}$ as compared to control untreated cells, it did not exert any cytotoxicity (almost nil) even at higher dose 200 $\mu\text{g}/\text{mL}$. Again at 400 $\mu\text{g}/\text{mL}$ dose, it exhibited similar slight toxicity as observed for 50 $\mu\text{g}/\text{mL}$ dose. So the result is not consistent as per the dosing.

It seems that all the NPs (60 nm, 120 nm and 250 nm) are showing very slight toxicity, as compared to untreated control cells at a broad concentration range (50-400 $\mu\text{g}/\text{mL}$). However, there is no dose dependent consistent result for 60 nm NPs, from which it can be interpreted that 60 nm NPs are more toxic than others. e.g. at 200 $\mu\text{g}/\text{mL}$ dose, 250 nm NPs are more toxic than 60 nm NPs and at 400 $\mu\text{g}/\text{mL}$ dose, the toxicity for 60 nm and 250 nm NPs are similar. Therefore, the interpretations of further study results which are based on the cell viability data are also misleading.

viii) The cytotoxic potential of 60 nm NPs at 200 $\mu\text{g}/\text{mL}$ is lower than that of 50 and 100 $\mu\text{g}/\text{mL}$ dose. Why did it happen?

ix) Why did you check MDA, SOD and ROS levels and what is the outcome of these data with respect to the cytotoxicity potential of NPs? It should be clearly explained in corresponding result sections.

x) How did the self-regulation of cells influence the higher SOD level for the treatment with 60 nm NPs, unlike other NPs?

xi) Why is the ROS level gradually reduced with increasing concentration of 60 nm NPs? The result is absurd.

xii) The text within Figure 8 is not visible. It should be presented in a proper way. Apoptosis cell % is not visible in the diagram and histogram.

xiii) Apoptosis data does not corroborate with the cell viability data. Cell viability shows almost biocompatible nature of 120 nm NPs except at 50 $\mu\text{g}/\text{mL}$ dose. But apoptosis study exhibited higher % apoptosis for 120 nm NPs treatment at all concentrations as compared to control untreated cells.

Why is there reduction in % apoptosis with increasing concentration of 120 nm NPs?

xiv) The English should be more polished.

xv) Conclusion section is poorly written.

Reviewer: 2

Comments to the Author(s)

The manuscripts reveals an interesting study of cytotoxicity of Fe_3O_4 nano particles on chicken macrophage cells to assess its effectiveness again *S. Entreditis* infection in poultry.

The report reveals that cytotoxicity of 250 nm particles is least and the particle of this size may be used as an antibiotic alternative. However, I have some concerns if such big size nanoparticles

can be used in drug delivery and authors must add some literature reports wherein particles greater than 200 nm has been used. Authors should also convince readers about the effectiveness of the 250 nm Fe₃O₄ nanoparticles to cross blood brain barrier since mostly for the drug delivery nanoparticle size used is between 10-100 nm.

Reviewer: 3

Comments to the Author(s)

Manuscript ID: RSOS-191561

Title: Cytotoxicity studies of Fe₃O₄ nanoparticles in chicken macrophage cells

Authors: Zhang, S., et al.

This manuscript describes the preparation of Magnetic Fe₃O₄ nanoparticles and examines them for their activity (oxidative index (malondialdehyde, superoxide dismutase and reactive oxygen species formation), apoptosis and pro-inflammatory cytokine secretion level in chicken macrophage cells as a function of nanoparticle size. They find that, when there are differences, there is greater activity in smaller nanoparticles.

The authors describe research that utilized magnetic nanoparticles of 60, 120 and 250 nm diameters. They provide a general approach for how the nanoparticles were prepared but not how the approach was used to get just 60 or 120 or 250 nm particles. This point is fairly important in the event that someone wants to reproduce the authors work.

The authors note that toxicity of nanoparticles should be examined prior to their use in vivo (e.g., p 7, lines 29-31, 'Therefore, it is important to fully investigate the biosafety of nanoparticles before they are used in biomedical or livestock production.' What is the fate of the nanoparticles? This is relevant here because if they are retained and the livestock eaten, they may also affect humans and so toxic effects in humans also needs to be examined.

The expected use of these iron nanoparticles should be stated. The authors conclude the manuscript with the statement 'Based on this study, 250 nm Fe₃O₄-NPs have good biological safety and maybe be used in disease diagnosis and substitution research in poultry industry.' (p. 9, conclusion section, lines 39-41). Are the authors suggesting the nanoparticles are going to be used to treat infections and take advantage of the ROS generation or, will the particles simply act as carriers. In the latter case, the particles studied in this manuscript would not be relevant as there surfaces were unmodified. The authors, at a minimum, need to describe how the nanoparticles are expected to be used. They may also want to state that in vivo degradation of particles with coatings are bearing 'payloads' (e.g., drugs) may result in the bare magnetic nanoparticles.

Misc:

Abstract: The sentence 'The results revealed that the cell viability of the 250 nm Fe₃O₄-NPs group was significantly greater ($P < 0.05$) than that of the 60 nm and 120 nm groups at a concentration of 50 μ g/mL, and the activity level of the 120 nm and 250 nm groups was significantly greater ($P < 0.01$) than that of the 60 nm group at a concentration of 100 μ g/mL.' is poorly written/confusing. If I understand it correctly, the authors are simply providing a rank order and could say that the smaller the particle size the greater the activity. If they feel it necessary, the qualifiers regarding concentration, etc. can be added.

The authors make some statements for which citations should be added p. 3, (Introduction section), lines 41-43, it is stated that 'There are few reports on the application of Fe₃O₄-NPs in animal husbandry.' If there are any reports, a representative citation(s) should be provided.

p. 3, lines 45-47 reads 'Our previous research found that Fe₃O₄-NPs could effectively control S. Enteritidis infection in chicken LMH cells.' This statement should be cited. If not published, then it should say 'unpublished research.'

p. 4, (under Materials and methods) line 21 reads 'fully dissolving in glycol...' should read fully dissolved in glycol...' Also, is this ethylene glycol (vs propylene glycol, etc.)?

Author's Response to Decision Letter for (RSOS-191561.R0)

See Appendix A.

RSOS-191561.R1 (Revision)

Review form: Reviewer 3

Is the manuscript scientifically sound in its present form?

Yes

Are the interpretations and conclusions justified by the results?

Yes

Is the language acceptable?

Yes

Do you have any ethical concerns with this paper?

No

Have you any concerns about statistical analyses in this paper?

No

Recommendation?

Accept as is

Comments to the Author(s)

This is a revised manuscript. The authors have adequately addressed my concerns and, in my opinion, the manuscript is suitable for publishing in its present form.

Decision letter (RSOS-191561.R1)

06-Mar-2020

Dear Miss Zhang:

Title: Cytotoxicity studies of Fe₃O₄ nanoparticles in chicken macrophage cells
Manuscript ID: RSOS-191561.R1

It is a pleasure to accept your manuscript in its current form for publication in Royal Society Open Science. The chemistry content of Royal Society Open Science is published in collaboration with the Royal Society of Chemistry.

RSC Associate Editor:
Comments to the Author:
(There are no comments.)

RSC Subject Editor:
Comments to the Author:
(There are no comments.)

Reviewer(s)' Comments to Author:
Reviewer: 3

Comments to the Author(s)

This is a revised manuscript manuscript. The authors have adequately addressed my concerns and, in my opinion, the manuscript is suitable for publishing in its present form.

Appendix A

中国农业科学院家禽研究所
Poultry Institute, Chinese Academy of Agricultural Sciences
江苏省家禽科学研究所
Jiangsu Institute of Poultry Science

No. 58 Cangjie Road
Yangzhou, 225125
P. R. China

Re: Version

Date: February 2020

Manuscript ID: RSOS-191561

TITLE: Cytotoxicity studies of Fe₃O₄ nanoparticles in chicken macrophage cells

Dear Editors,

Thank you very much for your hard work and valuable comments on our paper (Manuscript ID: RSOS-191561). We feel lucky that our manuscript went to these reviewers, as their comments not only helped us with the improvement of our manuscript but also offered some unique ideas for further studies. We have studied the comments carefully and have made corrections that we hope will be met with approval.

Please find below details regarding all of the changes we have made to the manuscript in response to the editor's and reviewers' comments. All of the changes are marked in red in both this response letter and the revised manuscript.

Reply to Reviewer 1:

i) Explanation of all the data is poor.

Reply: Thank you for your insightful suggestion. We have reinterpreted all of the data in parts 3.2, 3.3 and 3.4. (Part 3.2, "As shown in Figure 4, the cell viability of the 250 nm Fe₃O₄-NPs group was significantly greater than those of the 60 nm and 120 nm groups at the concentration of 50 µg/mL, and the activity levels of the 120 nm and 250 nm groups were significantly greater than that of the 60 nm group at a concentration of 100 µg/mL."; Part 3.3, "SOD levels of the 60 nm group were significantly higher than those of the 120 nm and 250 nm groups at 50 µg/mL. SOD levels of the 60 nm group were significantly higher, showing extreme differences, than those of the 120 nm and 250 nm groups at 100, 200 and 400 µg/mL."; Part 3.4, "At concentrations of 50 and 100 µg/mL, the

extent of apoptosis of the 250 nm group was significantly less than those of the 60 nm and 120 nm groups. At a concentration of 200 µg/mL, the extent of apoptosis of the 120 nm group was significantly less than that of the 60 nm group, and the extent of apoptosis of the 250 nm group was extremely significantly less than that of the 60 nm group. Additionally, the extent of apoptosis of the 250 nm group was significantly lower than that of the 60 nm group at 400 µg/mL.”)

ii) ‘...there are still some controversial results concerning the cytotoxic effects from use of SPIONs...’

The controversial results should be described in Introduction section.

Reply: Thank you for your insightful suggestion. Revisions have been made according to this suggestion in the introduction section. (Introduction section, “Some studies have reported that SPIONs are biologically benign [3], whereas other researchers believe that SPIONs have potential toxicity including organ toxicity and genotoxicity [1].”)

1. Liu G, Gao J, Ai H, Chen X. 2013 Applications and potential toxicity of magnetic iron oxide nanoparticles. *Small* 9, 1533-1545. (doi: 10.1002/sml.201201531)

3. Ivask A, Titma T, Visnapuu M, Vija H, Kakinen A, Sihtmae M, Pokhrel S, Madler L, Heinlaan M, Kisand V, Shimmo R, Kahru A. 2015 Toxicity of 11 Metal Oxide Nanoparticles to Three Mammalian Cell Types In Vitro. *Curr. Top. Med. Chem.* 15, 1914-1929. (doi: 10.2174/1568026615666150506150109)

iii) The synthesis procedure did not clarify the conditions of formation of three different sized (60 nm, 120 nm, 250 nm) nanoparticles.

Reply: Thank you for your insightful suggestion. Revisions have been made according to this suggestion in part 2.1. (Part 2.1, “Fe₃O₄-NPs with diameters of approximately 60 nm, 120 nm and 250 nm were prepared by a hydrothermal method with FeCl₃ and NaAc·3H₂O as raw materials. A 0.4 g or 0.6 g sample of FeCl₃ and 3.6 g of NaAc·3H₂O were added into the mixed solvent of 10 mL of glycol and 30 mL of diethylene glycol and stirred until fully dissolved by ultrasound, and these solutions were used to prepare 60 nm or 120 nm Fe₃O₄-NPs. A 0.82 g sample of FeCl₃ and 3.6 g of NaAc·3H₂O were added into 40 mL of glycol and stirred to achieve full dissolution by ultrasound, which was used to prepare the 250 nm Fe₃O₄-NPs. After complete dissolution, the mixture was transferred into a 50 mL Teflon-sealed autoclave and heated at 200 °C for 12 h.”)

iv) At which wavelength the absorbance of MDA/SOD was measured using ELISA? It should be

mentioned in experimental section.

Reply: Thank you for your insightful suggestion. Revisions have been made according to this suggestion in part 2.4. (Part 2.4, “The absorbance of MDA/SOD was measured at 532 nm/450 nm with a microplate reader (Infinite M200 Pro, Tecan, Switzerland).”)

v) Fluorescence microscopy or confocal microscopy would be more convincing for determination of ROS. The present work lacks of state-of-the art techniques.

Reply: Thank you for your insightful suggestion. In view of the actual situation and related literature [1-3], it is also feasible to use the reactive oxygen species assay kit to detect ROS.

1. Li KG, Chen JT, Bai SS, Wen X, Song SY, Yu Q, Li J, Wang YQ. 2009 Intracellular oxidative stress and cadmium ions release induce cytotoxicity of unmodified cadmium sulfide quantum dots. *Toxicol. In Vitro* **23**, 1007-1013. (doi: 10.1016/j.tiv.2009.06.020)
2. Jeelani R, Khan SN, Shaeib F, Kohan-Ghadr HR, Aldhaheeri SR, Najafi T, Thakur M, Morris R, Abu-Soud HM. 2017 Cyclophosphamide and acrolein induced oxidative stress leading to deterioration of metaphase II mouse oocyte quality. *Free Radic Blo. Med.* **110**, 11-18. (10.1016/j.freeradbiomed.2017.05.006)
3. Gong W, Li J, Zhu G, Wang Y, Zheng G, Kan Q. 2019 Chlorogenic acid relieved oxidative stress injury in retinal ganglion cells through IncRNA-TUG1/Nrf2. *Cell Cycle (Georgetown, Tex.)* **18**, 1549-1559. (doi: 10.1080/15384101.2019.1612697)

vi) Why is there no much difference in size of NPs obtained from TEM and DLS, especially for 120 nm and 250 nm NPs? Generally, the hydrodynamic size of NPs obtained in DLS is much greater than the size of NPs obtained in TEM.

Reply: Thank you for your insightful suggestion. There was not much difference in the size of NPs obtained from TEM and DLS, which may be because there was no modified molecules on the surface of these NPs and these NPs have good rigidity. Additionally, similar literature shows that the results of DLS and TEM show similar sizes [1].

1. Chen S, Chen S, Zeng Y, Lin L, Wu C, Ke Y, Liu G. 2018 Size-dependent superparamagnetic iron oxide nanoparticles dictate interleukin-1beta release from mouse bone marrow-derived macrophages. *J. Appl. toxicol.* **38**, 978-986. (doi: 10.1002/jat.3606)

vii) The interpretation of cell viability data (Figure 4) and the execution of the experiment are faulty. Although 60 nm NPs exerted slight cytotoxicity in HD11 cells at 50-100 µg/mL as compared to control untreated cells, it did not exert any cytotoxicity (almost nil) even at higher dose 200 µg/mL. Again at 400 µg/mL dose, it exhibited similar slight toxicity as observed for 50 µg/mL dose. So the result is not consistent as per the dosing.

It seems that all the NPs (60 nm, 120 nm and 250 nm) are showing very slight toxicity, as compared to untreated control cells at a broad concentration range (50-400 µg/mL). However, there is no dose dependent consistent result for 60 nm NPs, from which it can be interpreted that 60 nm NPs are more toxic than others. e.g. at 200 µg/mL dose, 250 nm NPs are more toxic than 60 nm NPs and at 400 µg/mL dose, the toxicity for 60 nm and 250 nm NPs are similar. Therefore, the interpretations of further study results which are based on the cell viability data are also misleading.

Reply: Thank you for your insightful suggestion. The main purpose of this study was to evaluate the cytotoxic effects of different sizes of Fe₃O₄-NPs in HD11 cells, and we analysed the cytotoxic effects of different sizes of Fe₃O₄-NPs on cell activity at the same concentration with the analysis of LDS's one-way analysis of variance (ANOVA). Since I did not describe this clearly in the previous paper, the data description has now been corrected in parts 3.2, 3.3 and 3.4 (same as the answer to the first question).

viii) The cytotoxic potential of 60 nm NPs at 200 µg/mL is lower than that of 50 and 100 µg/mL dose. Why did it happen?

Reply: Thank you for your insightful suggestion. We analysed the cytotoxic effects of different sizes of Fe₃O₄-NPs on the cell activity at the same concentration, and the cytotoxic effects of different concentrations (50, 100, 200 and 400 µg/mL) of 60 nm Fe₃O₄-NPs on cell activity were not significant ($P > 0.05$). Additionally, since I did not describe this clearly in the previous paper, the data description has now been corrected in part 3.2 (same as the answer to the first question).

ix) Why did you check MDA, SOD and ROS levels and what is the outcome of these data with respect to the cytotoxicity potential of NPs? It should be clearly explained in corresponding result sections.

Reply: Thank you for your insightful suggestion. By referring to the articles about the cytotoxicity of Fe₃O₄-NPs, we selected these indicators (MDA, SOD and ROS) for detection and repeated verification in chicken HD11 cells [1-3]. Relevant explanations have been added according to this suggestion in part 3.3. ("MDA is the final decomposition product of membrane lipid peroxidation

and widely used as a marker of oxidative lipid injury. SOD is a natural scavenger of oxygen free radicals in organisms, which can eliminate ROS and inhibit the harmful effects of oxidant molecules on tissues and cells. ROS produced in biological systems play an important role in cell damage, cell apoptosis and other metabolic activities.”) The results of our study showed that the apoptosis induced by Fe₃O₄-NPs was mainly related to oxidative stress due to increased ROS production.

1. Sarkar A, Sil PC. 2014 Iron oxide nanoparticles mediated cytotoxicity via PI3K/AKT pathway: role of quercetin. *Food Chem. Toxicol.* 71, 106-115. (doi: 10.1016/j.fct.2014.06.003)
2. Lee JH, Ju JE, Kim BI, Pak PJ, Choi EK, Lee HS, Chung N. 2014 Rod-shaped iron oxide nanoparticles are more toxic than sphere-shaped nanoparticles to murine macrophage cells. *Environ. Toxicol. Chem.* 33, 2759-2766. (doi: 10.1002/etc.2735)
3. Lin X, Zhao S, Zhang L, Hu G, Sun Z, Yang W. 2012 Dose-dependent Cytotoxicity and Oxidative Stress Induced by “Naked” FeO Nanoparticles in Human Hepatocyte. *Chem. Res. Chinese Universities* 28,114-118.

x) How did the self-regulation of cells influence the higher SOD level for the treatment with 60 nm NPs, unlike other NPs?

Reply: Thank you for your insightful suggestion. SOD plays an important role in limiting the damage caused by ROS. The ROS level for treatment with 60 nm Fe₃O₄-NPs was the highest among the ROS levels measured in this study, which may stimulate SOD activity in response to increased oxidative stress.

xi) Why is the ROS level gradually reduced with increasing concentration of 60 nm NPs? The result is absurd.

Reply: Thank you for your insightful suggestion. In this study, it can be seen that "the ROS level gradually decreased with increasing concentrations of 60 nm Fe₃O₄-NPs". At the same time, we should also notice that "the SOD level gradually increased with increasing concentrations of 60 nm Fe₃O₄-NPs". SOD is a scavenger of oxygen free radicals. With increasing concentrations of 60 nm NPs, the self-regulation of cells may stimulate SOD activity in response to increased oxidative stress by ROS.

xii) The text within Figure 8 is not visible. It should be presented in a proper way. Apoptosis cell %

is not visible in the diagram and histogram.

Reply: Thank you for your insightful suggestion. I will upload Figure 8 again in a more appropriate manner. The percentage of apoptotic cells is visible in the histogram.

xiii) Apoptosis data does not corroborate with the cell viability data. Cell viability shows almost biocompatible nature of 120 nm NPs except at 50 $\mu\text{g/mL}$ dose. But apoptosis study exhibited higher % apoptosis for 120 nm NPs treatment at all concentrations as compared to control untreated cells.

Why is there reduction in % apoptosis with increasing concentration of 120 nm NPs?

Reply: Thank you for your insightful suggestion. We analysed the cytotoxic effects of different sizes of Fe_3O_4 -NPs on cell activity and apoptosis at the same concentration. Additionally, because I did not describe this clearly in the previous version, the data description has now been corrected in parts 3.2, 3.3 and 3.4 (same as the answer to the first question)

xiv) The English should be more polished.

Reply: Thank you for your insightful suggestion. The editors at AJE have polished our manuscript. And the editing certificate is as follows.

xv) Conclusion section is poorly written.

Reply: The cell viability, oxidation index (MDA, SOD and ROS) and apoptosis level were detected

to analyse the cytotoxicity of Fe₃O₄-NPs of different sizes in HD11 cells. By referring to the articles about the cytotoxicity of Fe₃O₄-NPs, we selected these indicators for detection and repeated verification in chicken HD11 cells, and thus, we obtained the conclusion in this paper. Through reinterpretation of all the data in part 3, the conclusion of this study was obvious. The conclusion was that the smaller Fe₃O₄-NPs produced stronger cytotoxicity in chicken macrophage cells, which may be related to the oxidative stress and apoptosis induced by increased ROS production as well as the increased expression of pro-inflammatory cytokines. The 250 nm Fe₃O₄-NPs have good biological safety and may be used in disease diagnosis and substitution research in the poultry industry.

Reply to Reviewer 2:

The report reveals that cytotoxicity of 250 nm particles is least and the particle of this size may be used as an antibiotic alternative. However, I have some concerns if such big size nanoparticles can be used in drug delivery and authors must add some literature reports wherein particles greater than 200 nm has been used. Authors should also convince readers about the effectiveness of the 250 nm Fe₃O₄ nanoparticles to cross blood brain barrier since mostly for the drug delivery nanoparticle size used is between 10-100 nm.

Reply: Thank you for your insightful suggestion. In this study, 250 nm Fe₃O₄-NPs have good biological safety, which lays a foundation for the potential application of Fe₃O₄-NPs in the poultry industry. However, Fe₃O₄-NPs as potential antibiotic substitutes and their mechanism require further experiments. Shi *et al.* reported that the 200 nm Fe₃O₄-NPs may be a potential antibiotic alternative to control *Salmonella enteritidis* (*S. enteritidis*) infection during clinical therapy and in poultry industry operations [1]. Our previous research found that Fe₃O₄-NPs could be observed in the livers of chickens, indicating that they could pass through the blood brain barrier [2].

1. Shi S, Wu S, Shen Y, Zhang S, Xiao Y, He X, Gong J, Farnell Y, Tang Y, Huang Y, Gao L. 2018 Iron oxide nanozyme suppresses intracellular Salmonella Enteritidis growth and alleviates infection in vivo. *Theranostics* **8**, 6149-6162. (doi: [10.7150/thno.29303](https://doi.org/10.7150/thno.29303))
2. Shen Y, Xiao Y, Zhang S, Wu S, Gao L, Shi S. 2019 Fe₃O₄ Nanoparticles Attenuated Salmonella Infection in Chicken Liver Through Reactive Oxygen and Autophagy via PI3K/Akt/mTOR Signaling. *Front. Physiol.* **10**, 1580. (doi: [10.3389/fphys.2019.01580](https://doi.org/10.3389/fphys.2019.01580))

Reply to Reviewer 3:

The authors describe research that utilized magnetic nanoparticles of 60, 120 and 250 nm diameters. They provide a general approach for how the nanoparticles were prepared but not how the approach was used to get just 60 or 120 or 250 nm particles. This point is fairly important in the event that someone wants to reproduce the authors work.

Reply: Thank you for your insightful suggestion. Revisions have been made according to this suggestion in part 2.1. (Part 2.1, “Fe₃O₄-NPs with diameters of approximately 60 nm, 120 nm and 250 nm were prepared by a hydrothermal method with FeCl₃ and NaAc·3H₂O as raw materials. A 0.4 g or 0.6 g sample of FeCl₃ and 3.6 g of NaAc·3H₂O were added into the mixed solvent of 10 mL of glycol and 30 mL of diethylene glycol and stirred until fully dissolved by ultrasound, and these solutions were used to prepare 60 nm or 120 nm Fe₃O₄-NPs. A 0.82 g sample of FeCl₃ and 3.6 g of NaAc·3H₂O were added into 40 mL of glycol and stirred to achieve full dissolution by ultrasound, which was used to prepare the 250 nm Fe₃O₄-NPs. After complete dissolution, the mixture was transferred into a 50 mL Teflon-sealed autoclave and heated at 200 °C for 12 h.”)

The authors note that toxicity of nanoparticles should be examined prior to their use in vivo (e.g., p 7, lines 29-31, ‘Therefore, it is important to fully investigate the biosafety of nanoparticles before they are used in biomedical or livestock production.’ What is the fate of the nanoparticles? This is relevant here because if they are retained and the livestock eaten, they may also affect humans and so toxic effects in humans also needs to be examined.

Reply: Thank you for your insightful suggestion. There are some reports on the cytotoxicity of Fe₃O₄ nanoparticles in mammals (rats, mice) or humans [1-4]. Poultry has many different characteristics in structure and function from mammals, and HD11 cells share many similarities with normal chicken macrophages and have the advantages of rapid subculture growth; hence, they can be used as an ideal model of chicken macrophages for in vitro studies [5-7]. In this paper, we mainly evaluated the cytotoxicity of Fe₃O₄-NPs in HD11 cells at the cellular level in order to select the relatively safe size of Fe₃O₄-NPs. In addition, we will carry out animal experiments to verify the safety and residual problems of Fe₃O₄-NPs. Thank you for your understanding.

1. Lee JH, Ju JE, Kim BI, Pak PJ, Choi EK, Lee HS, Chung N. 2014 Rod-shaped iron oxide nanoparticles are more toxic than sphere-shaped nanoparticles to murine macrophage cells. *Environ. Toxicol. Chem.* **33**, 2759-2766. (doi: 10.1002/etc.2735)

2. Chen S, Chen S, Zeng Y, Lin L, Wu C, Ke Y, Liu G. 2018 Size-dependent superparamagnetic iron oxide nanoparticles dictate interleukin-1 β release from mouse bone marrow-derived macrophages. *J. Appl. toxicol.* **38**, 978-986. (doi: 10.1002/jat.3606)
3. Ge G, Wu H, Xiong F, Zhang Y, Guo Z, Bian Z, Xu J, Gu C, Gu N, Chen X, Yang D. 2013 The cytotoxicity evaluation of magnetic iron oxide nanoparticles on human aortic endothelial cells. *Nanoscale Res. Lett.* **8**, 215. (doi: 10.1186/1556-276X-8-215)
4. Ankamwar B, Lai TC, Huang JH, Liu RS, Hsiao M, Chen CH, Hwu YK. 2010 Biocompatibility of Fe₃O₄ nanoparticles evaluated by in vitro cytotoxicity assays using normal, glia and breast cancer cells. *Nanotechnology* **21**, 75102. (doi: 10.1088/0957-4484/21/7/075102)
5. Xie H, Raybourne RB, Babu US, Lillehoj HS, Heckert RA. 2003 CpG-induced immunomodulation and intracellular bacterial killing in a chicken macrophage cell line. *Dev. Comp. Immunol.* **27**, 823-834. (doi: 10.1016/s0145-305x(03)00079-x)
6. He H, Genovese KJ, Nisbet DJ, Kogut MH. 2008 Phospholipase C, phosphatidylinositol 3-kinase, and intracellular [Ca²⁺] mediate the activation of chicken HD11 macrophage cells by CpG oligodeoxynucleotide. *Dev. Comp. Immunol.* **32**, 1111-1118. (doi: 10.1016/j.dci.2008.02.009)
7. Babu US, Sommers K, Harrison LM, Balan KV. 2012 Effects of fructooligosaccharide-inulin on Salmonella-killing and inflammatory gene expression in chicken macrophages. *Vet. Immunol. Immunopathol.* **149**, 92-96. (doi: 10.1016/j.vetimm.2012.05.003)

The expected use of these iron nanoparticles should be stated. The authors conclude the manuscript with the statement ‘Based on this study, 250 nm Fe₃O₄-NPs have good biological safety and maybe be used in disease diagnosis and substitution research in poultry industry.’ (p. 9, conclusion section, lines 39-41). Are the authors suggesting the nanoparticles are going to be used to treat infections and take advantage of the ROS generation or, will the particles simply act as carriers. In the latter case, the particles studied in this manuscript would not be relevant as their surfaces were unmodified. The authors, at a minimum, need to describe how the nanoparticles are expected to be used. They may also want to state that in vivo degradation of particles with coatings are bearing ‘payloads’ (e.g., drugs) may result in the bare magnetic nanoparticles.

Reply: Thank you for your insightful suggestion. In this paper, we mainly evaluated the cytotoxicity of Fe₃O₄-NPs in chicken macrophage cells in order to select the appropriate size of Fe₃O₄-NPs for the next test. In the next study, we will establish an *S. enteritidis*-infected chicken macrophage model to study the effects of Fe₃O₄-NPs as potential antibiotic substitutes for *S.*

enteritidis-infected chicken macrophages and their mechanism. Revisions have been made according to this suggestion in the conclusion. (Part 4, “Based on this study, 250 nm Fe₃O₄-NPs have good biological safety, which lays a foundation for the potential application of Fe₃O₄-NPs in the poultry industry. The study of Fe₃O₄-NPs as potential antibiotic substitutes and their mechanism requires further experiments.”)

Misc:

Abstract: The sentence ‘The results revealed that the cell viability of the 250 nm Fe₃O₄-NPs group was significantly greater (P < 0.05) than that of the 60 nm and 120 nm groups at a concentration of 50 µg/mL, and the activity level of the 120 nm and 250 nm groups was significantly greater (P < 0.01) than that of the 60 nm group at a concentration of 100 µg/mL.’ is poorly written/confusing. If I understand it correctly, the authors are simply providing a rank order and could say that the smaller the particle size the greater the activity. If they feel it necessary, the qualifiers regarding concentration, etc. can be added.

Reply: Thank you for your insightful suggestion. Revisions have been made according to this suggestion in the abstract section. (Abstract section, “The results revealed that the cell viability of the 60 nm Fe₃O₄-NPs group was lower than those of the 120 nm and 250 nm groups when the same concentration of Fe₃O₄-NPs was added.”)

The authors make some statements for which citations should be added

p. 3, (Introduction section), lines 41-43, it is stated that ‘There are few reports on the application of Fe₃O₄-NPs in animal husbandry.’ If there are any reports, a representative citation(s) should be provided.

Reply: Thank you for your insightful suggestion. The references have been added. (“There are few reports on the application of Fe₃O₄-NPs in animal husbandry [17,18].”)

17. Nikonov IN, Folmanis YG, Folmanis GE, Kovalenko LV, Laptev GY, Egorov IA, Fisinin VI, Tananaev IG. 2011 Iron nanoparticles as a food additive for poultry. *Dokl. Biol. Sci.* **440**, 328-331. (doi: 10.1134/S0012496611050188)

18. Rehman H, Akram M, Kiyani MM, Yaseen T, Ghani A, Saggiu JI, Shah S, Khalid ZM, Bokhari S. 2020 Effect of Endoxylanase and Iron Oxide Nanoparticles on Performance and Histopathological Features in Broilers. *Biol. Trace Elem. Res.* **193**, 524-535. (doi: 10.1007/s12011-019-01737-z)

p. 3, lines 45-47 reads ‘Our previous research found that Fe₃O₄-NPs could effectively control *S. Enteritidis* infection in chicken LMH cells.’ This statement should be cited. If not published, then it should say ‘unpublished research.’

Reply: Thank you for your insightful suggestion. The reference has been added. (“Our previous research found that Fe₃O₄-NPs could effectively control *S. enteritidis* infection in chicken LMH cells [11].”)

11. Shi S, Wu S, Shen Y, Zhang S, Xiao Y, He X, Gong J, Farnell Y, Tang Y, Huang Y, Gao L. 2018 Iron oxide nanozyme suppresses intracellular *Salmonella* Enteritidis growth and alleviates infection in vivo. *Theranostics* **8**, 6149-6162. (doi: 10.7150/thno.29303)

p. 4, (under Materials and methods) line 21 reads ‘fully dissolving in glycol...’ should read fully dissolved in glycol...’ Also, is this ethylene glycol (vs propylene glycol, etc.)?

Reply: Thank you for your insightful suggestion. Revisions have been made according to this suggestion in part 2.1 (same as the answer to the first question).

Thank you for your patience in reading all of our responses. We hope that these responses and our revisions are acceptable. Please do not hesitate to contact me if additional revisions are required.

Yours sincerely,

Shan Zhang

Institute of Poultry Science of Jiangsu Province

Yangzhou 225125

P. R. China

Tel: +86-514-85599035

Email: zhangshan3321@163.com